

# Evaluation of a novel syndromic surveillance system for the detection of the 2007 melamine-related nephrotoxicosis foodborne outbreak in dogs and cats in the United States

Hsin-Yi Weng[1], Mark A.L. Gaona[2] and Philip H. Kass[3]

[1] Department of Comparative Pathobiology, Purdue University, West Lafayette, IN, USA
[2] Enterprise Student Applications Unit of Information and Educational Technology Department, University of California, Davis, Davis, CA, USA
[3] Department of Population Health and Reproduction, University of California, Davis, Davis, CA, USA

Corresponding author
Hsin-Yi Weng, weng9@purdue.edu

## ABSTRACT

The 2007 nephrotoxicosis outbreak associated with melamine and cyanuric acid adulteration of pet foods in the United States sparked an urgent need for a nationwide companion animal surveillance program. In 2016, we introduced a syndromic surveillance system based on a novel epidemiological algorithm, the proportionate diagnostic outcome ratio (PDOR). The PDOR procedure was validated using simulated outbreaks of foodborne illness (i.e., aflatoxicosis and gastrointestinal illness) in dogs and cats. In this study, we further evaluated the PDOR procedure using the 2007 melamine-related outbreak of nephrotoxicosis. The performance of the PDOR procedure was assessed by the time to alert and positive predictive value (PPV). Electronic medical records of dogs and cats seen at networked primary care veterinary hospitals across the United States were retrieved from a centralized database. The data of four relevant syndromic components: elevated serum creatinine concentration, vomiting, anorexia, and lethargy from July 28, 2006 to May 31, 2007 were prospectively analyzed using the PDOR algorithm. The results showed that the alerts generated from the analysis of elevated serum creatinine concentration could have led to an early detection of this nephrotoxicosis foodborne outbreak and were well matched to the reported timeline of the outbreak. Additionally, we also observed variations in the performance of the PDOR procedure across age of animals and syndromic components, with the PPVs ranged from 0.61 to 1.0. Combined with the findings from previous evaluations using simulated outbreak scenarios, this study provided additional evidence that the PDOR procedure can be applied in syndromic surveillance to effectively and accurately detect various types of foodborne illness outbreaks in companion animals. However, the interpretations of and responses to alerts require an understanding of clinical veterinary medicine and relevant syndromic knowledge, and should not be based solely on quantitative measures.

# INTRODUCTION

In late February and early March 2007 complaints of pet cats developing acute renal failure unveiled a large-scale foodborne illness outbreak in dogs and cats in the United States (*Puschner & Reimschuessel, 2011*). Further investigations confirmed that two chemicals, melamine and cyanuric acid, had been added to adulterate wheat gluten used in the manufacture of pet foods for fraudulently increasing the apparent protein content in those products (*Dobson et al., 2008*). Consumption of both melamine and cyanuric acid can form a large quantity of crystals in the lumens of renal tubules, which can cause extensive lesions in renal tubular epithelial cells and result in crystalluria, uroliths, and nephrotoxicosis (*Puschner & Reimschuessel, 2011*). On March 16, 2007 the pet food supplier, Menu Foods Inc., announced a voluntary recall of certain pet food brands in the United States. At the end of the crisis, more than 54 million cans or pouches of dog and cat food in the United States and Canada were recalled, marking one of the most costly food recalls in the U.S. history (*Rumbeiha & Morrison, 2011*). Although the full extent of the pet deaths and illnesses caused by the chemical adulterant is unknown, one survey reported at least 424 confirmed cases (66% cats and 34% dogs) (*Rumbeiha & Morrison, 2011*).

This large-scale outbreak underscores the necessity of a nationwide surveillance system to safeguard the health of companion animals. The great concerns expressed by the public following this outbreak led to significant changes in legislation on pet food process and safety. In response, the U.S. Food and Drug Administration (FDA) launched the Pet Food Early Warning Surveillance System, with the goal of quickly identifying outbreaks of pet-food associated illness (*U.S. Food & Drug Administration, 2019*). The two data resources are the FDA's Consumer Complaint Reporting System and the FDA-National Institutes of Health Safety Reporting Portal (https://www.safetyreporting.hhs.gov/), both of which are reliant on voluntary consumer reports and complaints. While this approach to surveillance might assist in early detection of potential pet-food associated problems, its accuracy and reliability rely heavily on the quality and extent of consumer reporting.

Our research team previously published the analytical and interpretive protocols for a syndromic surveillance system, Aberrant Diagnostic Outcome Repository in Epidemiology (ADORE), of companion animal health in the United States (*Kass et al., 2016*). ADORE utilizes a novel algorithm, the proportionate diagnostic outcome ratio (PDOR), which is modified from epidemiological measures, such as the proportional mortality ratio and proportional reporting ratio, to adapt to the analysis of clinical diagnostic test results (*Miettinen & Wang, 1981*; *Rothman, Lanes & Sacks, 2004*). A PDOR is computed as the ratio of the proportion of patients observed with a diagnostic outcome, utilizing time and/or geographic region as the exposures of interest. It differs from other statistical algorithms for signal detection such as control charts and moving average. Details on the

PDOR and ADORE scoring system are described in the Materials and Methods section. The performance of the PDOR algorithm has previously been evaluated using simulated outbreak scenarios of aflatoxicosis and gastrointestinal illness constructed by independent experts and that were blinded to the study investigators.

Several unique features of the PDOR algorithm are highlighted here. First, the PDOR procedure differs from traditional pattern-recognition algorithms, and is adapted from classical epidemiologic measures. Second, the five-tiered alert system of ADORE includes a built-in and modifiable scoring matrix that is based on the measures of association and precision instead of statistical significance. Third, the analyses are performed separately by species and age strata of patients, which can improve both sensitivity and specificity for aberration detection. Fourth, the algorithm implemented in the ADORE system is suitable for detecting a gradual event as well as an acute outbreak by providing flexibility to its users for selecting different baseline windows. Lastly, the analysis and reporting process is automated with a fast processing time to provide near real-time reports. Details on the methodology of the ADORE system can be found in our 2016 publication (*Kass et al., 2016*).

In this study, we further evaluated the performance of the PDOR procedure and the ADORE scoring system in detecting the 2007 nephrotoxicosis outbreak associated with melamine and cyanuric acid adulteration of pet foods using veterinary medical data from the largest networked primary care veterinary hospitals in the United States. We hypothesized that the PDOR procedure of the ADORE system would be able to detect an abnormal signal between December 3, 2006 and March 16, 2007, and the alerts would last beyond March 31, 2007. December 3, 2006 was the earliest purchase date of the problematic pet foods indicated in the initial recall announced on March 16, 2007. We also hypothesized that the sensitivity of the PDOR procedure would vary across species and age of animals, as well as syndromic components.

## MATERIALS AND METHODS

Medical records of canine and feline patients stored in a centralized database of the participating networked primary care veterinary hospitals between July 28, 2006 and May 31, 2007 were transferred to the ADORE system for analyses. During the study period, the database contained medical records from more than 750 primary care veterinary hospitals in 43 states of the United States. Each medical record contained information on the hospital visit of a patient on a particular day. One animal could have contributed multiple records on different days, but not on the same day. Four syndromic components (i.e., clinical presentations and a laboratory result) pertaining to nephrotoxicosis, including elevated serum creatinine concentration (dog: creatinine >1.4 mg/dl; cat: creatinine >1.6 mg/dl), vomiting, anorexia, and lethargy were included in this study.

The epidemiologic measure, PDOR, is the main algorithm for aberration detection implemented in the ADORE system. PDOR for a particular date $i$ is computed by dividing the diagnostic outcome proportion on date $i$ ($DOP_i$) by the baseline DOP ($DOP_B$), in which DOP is the number of syndromic cases divided by the number of medical records. Table 1 summarizes the equations for DOP, PDOR and confidence interval (CI).

 

**Table 1 Equations for diagnostic outcome proportion (DOP), proportionate diagnostic outcome ratio (PDOR), standard error (SE), and confidence interval (CI).**

| | |
|---|---|
| DOP | (Number of syndromic cases)/(Number of medical records) |
| PDOR | $\mathrm{DOP}_i / \mathrm{DOP}_B$ <br> where $\mathrm{DOP}_i$ and $\mathrm{DOP}_B$ are diagnostic outcome proportion on date $i$ and during baseline window, respectively |
| SE for ln(PDOR)[a] | $\sqrt{\dfrac{(1 - \mathrm{DOP}_i)}{C_i} + \dfrac{(1 - \mathrm{DOP}_B)}{C_B}}$ <br> where $C_i$ and $C_B$ are number of syndromic cases on date $i$ and during baseline window, respectively |
| CI | $e^{[ln(\mathrm{PDOR}) \pm k \times SE_{ln(\mathrm{PDOR})}]}$ <br> where $e$: exponential function; $k = 1.28$ for 80% CI and 1.96 for 95% CI |

**Note:**
[a] Natural logarithm of PDOR.

In this study, we used a 7-day baseline window and a 90-day lag time, with the specific objective of detecting a gradually increasing outbreak. Thus, for example, the PDOR on May 1, 2007 was computed as the DOP on May 1, 2007 divided by the $\mathrm{DOP}_B$ accumulated between January 26, 2006 and February 1, 2007. Because of the selection of a 90-day lag time, the data from July 28, 2006 to October 31, 2006 were used as the baseline and the data from November 1, 2006 to May 31, 2007 were actively investigated. Although medical records were retrospectively used in this study, the analyses were done prospectively by moving forward in time for each investigated date and its corresponding baseline window. To avoid using a baseline window that might be contaminated by an aberrant event, a baseline window that had a lower limit of the 80% CI for PDOR >1 for all seven days was replaced with the preceding window. Likewise, if the PDORs in a baseline window were all <0.8, it was replaced with the preceding window as well. Different lag times of 30-day, 60-day, and 120-days were investigated in sensitivity analyses.

The PDOR analyses were done separately by syndromic components ($N = 4$), species (canine and feline), and age group (all ages, <3 years, 3–7 years, 8–12 years, and 13 or more years). Thus, the ADORE system performed the PDOR procedure 40 times each day. After the PDOR procedure, an alert score was computed according to different outcome parameters, including: (1) total number of cases, (2) PDOR, (3) the lower bounds of an 80% CI and 95% CI for PDOR, and (4) whether there were alerts across different age groups, the investigated syndromes, or three or more days in a 7-day period (Table 2).

This summary score, ranging from 0 to 26, was then used to generate a five-tiered alert. The first tier alert (green; score <9) was normal, with successively higher tier 2 (blue; score 9 to <13), tier 3 (yellow; score 13 to <16), tier 4 (orange; score 16 to <19), and tier 5 (red; score ≥19) for the strongest indication of an outbreak. Detailed PDOR algorithms and the scoring matrix of the ADORE system can be found in a previously published study (Kass et al., 2016). To validate the PDOR procedure, we compared the alerts generated by the ADORE system against the reported timeline of the 2007 melamine-related nephrotoxicosis foodborne outbreak. The outbreak period was defined

**Table 2 Scoring matrix for alert score computation and classification.**

| Parameter | Score/classification |
|---|---|
| Number of cases | |
| <3 | 0 |
| ≥3 | 2 |
| PDOR[a] | |
| <1.25 | 0 |
| 1.25–1.99 | 3 |
| 2–2.99 | 5 |
| 3–3.99 | 7 |
| ≥4 | 9 |
| 80% LCL[b] | |
| ≤1.1 | 0 |
| >1.1 | 2 |
| 95% LCL[b] | |
| ≤0.8 | 0 |
| >0.8–1.1 | 2 |
| >1.1 | 4 |
| Across age groups | |
| Sum[c] ≥ 5 in 2 or more groups | 3 |
| Else | 0 |
| Across syndromes | |
| Sum[c] ≥ 5 in 2 or more syndromes | 2 |
| Else | 0 |
| Across days | |
| Sum[c] ≥ 6 on 3 or more days in a week | 4 |
| Else | 0 |

**Notes:**
[a] Proportionate diagnostic outcome ratio.
[b] Lower confidence limit (LCL) of a PDOR.
[c] Sum score of number of cases, PDOR, 80% LCL, and 95% LCL.

as December 3, 2006 to April 30, 2007 in this investigation. In addition, we examined the hypothesis whether the PDOR procedure of the ADORE system would be able to detect an abnormal signal between December 3, 2006 and March 16, 2007. Detection of a signal during this period represents an early detection compared with the official recall announcement on March 16, 2007. To quantify the magnitude of false alerts (i.e. alerts that are generated outside the defined outbreak period), we present two measures: (1) daily average number of alerts and (2) positive predictive value (PPV), defined as the proportion of alerts generated by the ADORE system that were correct (i.e., generated during the defined outbreak period). The algorithms implemented in the ADORE system allow for the detection of both temporal and spatial clusters. Because the contaminated pet foods were distributed and sold across the United States, resulting in a nationwide outbreak, we focused on the evolution of the PDOR's performance for temporal aberrant event detection in this study.
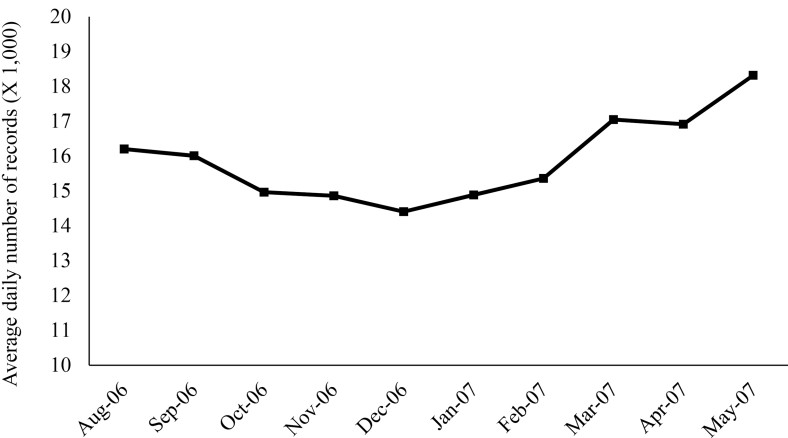

**Figure 1 Monthly pattern of the average daily number of medical records retrieved from a centralized database of the participating primary care veterinary hospital in the United States.** Each medical record represents the hospital visit of a patient on a particular day. One animal could have contributed multiple records on different days, but not on the same day.

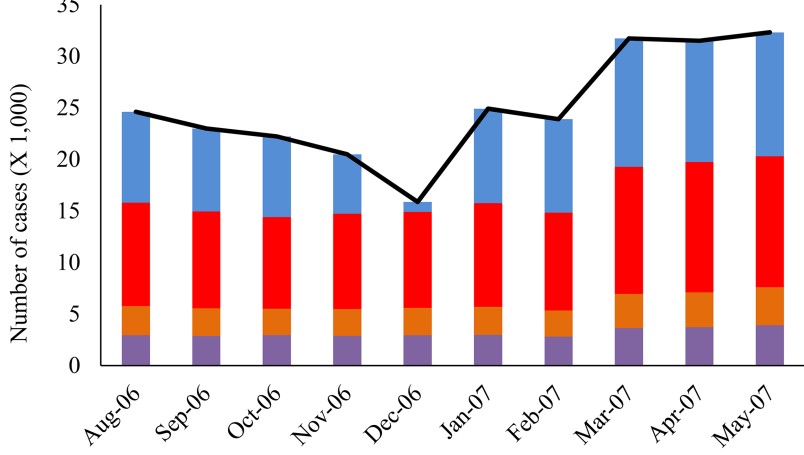

**Figure 2 Distribution of the number of cases by syndromic components and months.** Blue: Creatinine; red: Vomiting; orange: Lethargy; purple: Anorexia; black line: Total.

# RESULTS

During the study period ($N$ = 307 days), 4,895,123 veterinary medical records were processed by the ADORE system using the PDOR procedure, with the majority being canine patients (83%) and young adults less than 8 years old (83%). The average daily number of medical records was 15,945, which varied by weekdays and holidays. Although no monthly pattern in the average daily number of medical records was observed, the daily number of medical records increased from March 2007 to May 2007 (Fig. 1). Overall, similar patterns were also observed in the total number of cases across all investigated syndromic components (Fig. 2). The composition of the syndromic components showed that the observed patterns were mainly driven by elevated creatinine and vomiting (Fig. 2).

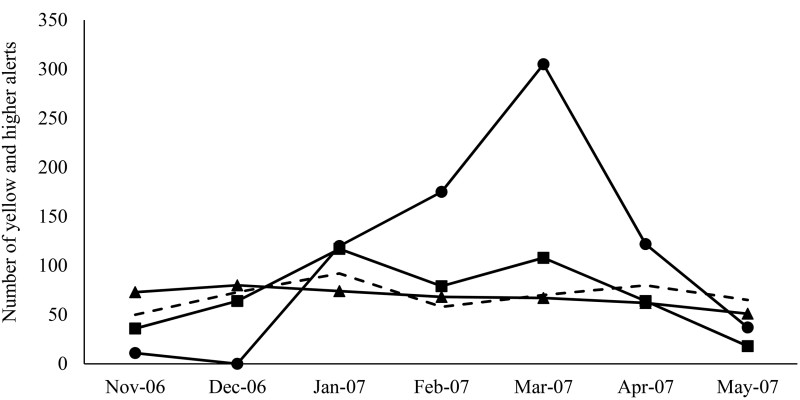

**Figure 3 Monthly distribution of the high tier alerts (yellow and higher) generated by the ADORE system for each syndromic component.** Solid line with the circle marks: Creatinine; square marks: Vomiting; triangle marks: Anorexia. Dashed line: Lethargy.

During the study period, 2,219 high-tiered alerts (i.e., yellow, orange, and red alerts) were generated. The total number of high-tiered alerts doubled in January 2007 compared to December 2006, and peaked in March 2007, with a substantial increase in the number of red alerts. Further breakdown showed that on average, 2.2 and 12.6 alerts at yellow or higher tiers were generated daily outside and within the outbreak period, respectively. The majority (86.4%) of the high-tiered alerts outside the outbreak period were from vomiting, anorexia, and lethargy, whereas the high-tiered alerts from elevated creatinine started in January 2007 and peaked in March 2007 (Fig. 3).

The results generated by the ADORE system indicated that elevated creatinine had the best performance in detecting the 2007 pet food-associated outbreak among the investigated syndromic components (Figs. 4A and 4B). With a 90-day lag time and 7-day baseline window, the alerts (blue and higher) started in early January 2007, peaked in March 2007, and lasted until May 2007. Similar patterns were observed in both dogs and cats, across age groups of <3 years old, 3 to <8 years old, and 8 to <13 years old. In cats, the same alert pattern was observed in the age group of ≥13 years old, whereas the alerts were more consistent starting in late February 2007 in dogs aged 13 years and older (Fig. 4C).

The alerts for vomiting, lethargy, and anorexia were less conclusive compared to the alerts for elevated creatinine (Figs. 5–7). Similar observations applied to all age groups, and thus only the results of all ages combined are reported. The 7-day moving average indicated that the alerts for vomiting elevated only at the blue tier between December 3, 2006 and March 16, 2007 in both dogs and cats with the peak occurring on and after March 21, 2007 (Fig. 5). Similar patterns were observed for lethargy and anorexia in dogs, with an exception for an early peak in late November 2006 for anorexia (Figs. 6A and 7A). In cats, multiple peaks at the blue and yellow tiers were observed in the alerts for lethargy and anorexia, respectively, between December 3, 2006 and March 16, 2007 (Figs. 6B and 7B).

The PPVs further support the superior performance of elevated serum creatinine concentration on generating true positive alerts compared with other syndromic

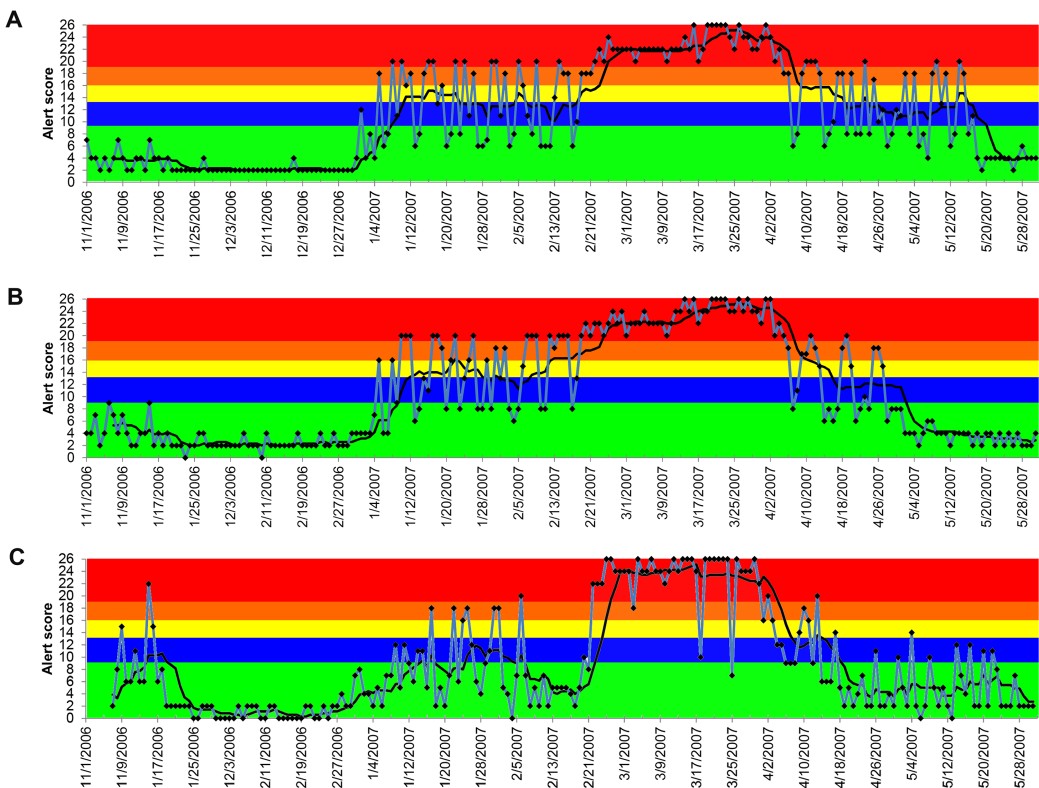

**Figure 4 Temporal distribution of the alert scores for elevated serum creatinine concentration generated by the ADORE system.** (A) Alert scores for dogs, all ages combined. (B) Alert scores for cats, all ages combined. (C) Alert scores for dogs 13 years and older. Blue line: daily alert score; black line: 7-day moving average of alert score; background colors represent alert tiers: green (alert score <9); blue (alert score 9 to <13); yellow (alert score 13 to <16); orange (alert score 16 to <19); red (alert score ≥19).

components (Table 3). No noticeable pattern in PPV across age groups within each animal species was observed. In contrast to age, PPVs in cats tended to be greater than PPVs in dogs across all syndromic components except for lethargy. The sensitivity analyses revealed an influence of baseline-window lag time on alert generation, but only for elevated creatinine (Table 4). The peak alert score varied depending on the lag time used, whereas the start date and end date shifted only slightly.

## DISCUSSION

The 2007 pet food recall due to melamine and cyanuric acid adulterations was the costliest pet food recall in the U.S. history. This and several other large-scale foodborne illness outbreaks in dogs and cats during the last decade signaled the need for a nationwide companion animal surveillance program (*Bischoff & Rumbeiha, 2018*; *Burns, 2007*). However, compared to advancements in computational technology and data science, the development of companion animal surveillance seems to be lagging (*McGreevy et al., 2017*; *Muellner et al., 2016*; *Hale et al., 2019*). Lack of centralized veterinary medical databases is probably one of the largest challenges hampering its progress (*O'Neill et al., 2014*). Our collaboration with the largest network of primary care veterinary hospitals in

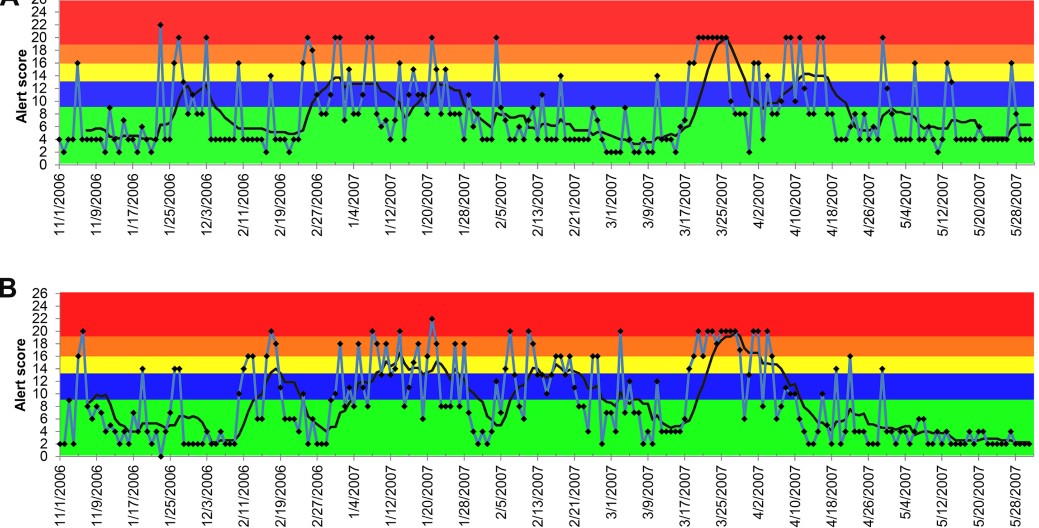

**Figure 5 Temporal distribution of the alert scores for vomiting generated by the ADORE system.**
(A) Alert scores for dogs, all ages combined. (B) Alert scores for cats, all ages combined. Blue line: daily alert score; black line: 7-day moving average of alert score; background colors represent alert tiers: green (alert score <9); blue (alert score 9 to <13); yellow (alert score 13 to <16); orange (alert score 16 to <19); red (alert score ≥19).

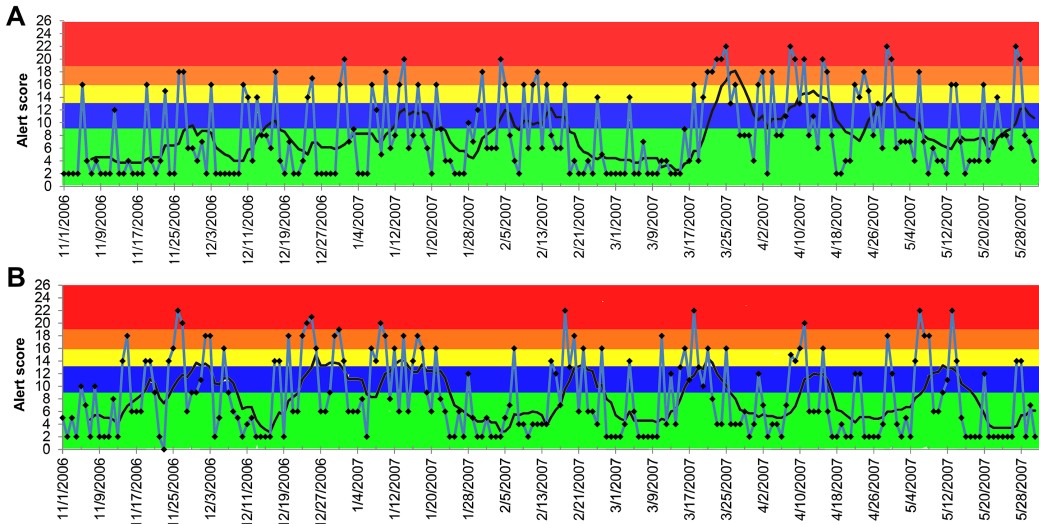

**Figure 6 Temporal distribution of the alert scores for lethargy generated by the ADORE system.**
(A) Alert scores for dogs, all ages combined. (B) Alert scores for cats, all ages combined. Blue line: daily alert score; black line: 7-day moving average of alert score; background colors represent alert tiers: green (alert score <9); blue (alert score 9 to <13); yellow (alert score 13 to <16); orange (alert score 16 to <19); red (alert score ≥19).

the United States made it possible for us to carry out this study. The electronic veterinary medical records of the animals seen at the participating hospitals are standardized and streamlined into a centralized database. This database provides tremendous opportunities for companion animal surveillance and a near-real time alerting system.

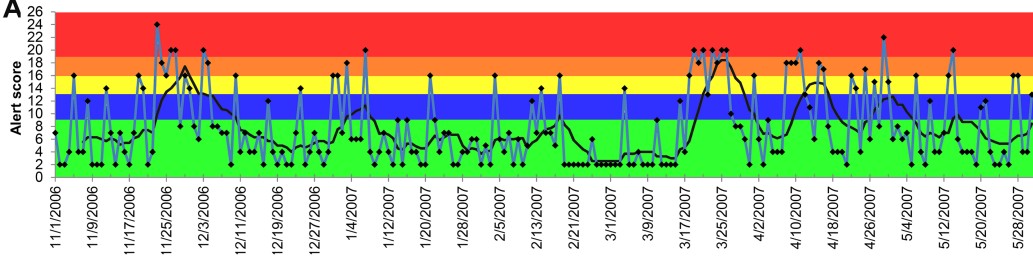

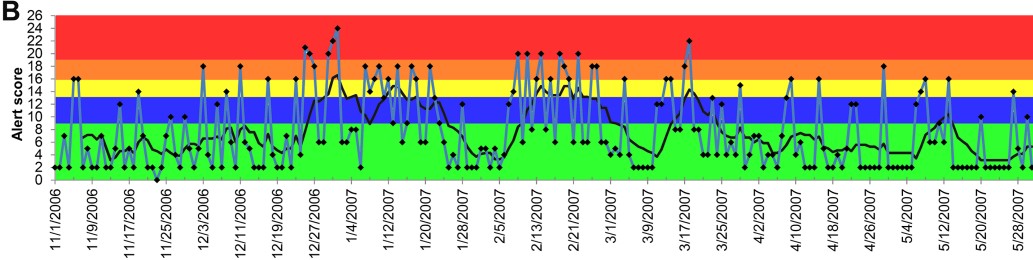

**Figure 7 Temporal distribution of the alert scores for anorexia generated by the ADORE system.**
(A) Alert scores for dogs, all ages combined. (B) Alert scores for cats, all ages combined. Blue line: daily alert score; black line: 7-day moving average of alert score; background colors represent alert tiers: green (alert score <9); blue (alert score 9 to <13); yellow (alert score 13 to <16); orange (alert score 16 to <19); red (alert score ≥19).

**Table 3 Proportion of alerts (i.e., true positives) that are generated during the study outbreak period from December 3, 2006 to April 30, 2007.**

|  | Creatinine | Vomiting | Anorexia | Lethargy |
|---|---|---|---|---|
| Dog |  |  |  |  |
| <3 | 0.83 | 0.71 | 0.66 | 0.66 |
| 3–8 | 0.91 | 0.86 | 0.68 | 0.85 |
| 8–13 | 0.94 | 0.93 | 0.79 | 0.75 |
| ≥13 | 0.85 | 0.76 | 0.70 | 0.80 |
| All ages combined | 0.90 | 0.83 | 0.69 | 0.82 |
| Cat |  |  |  |  |
| <3 | 0.92 | 0.93 | 0.74 | 0.72 |
| 3–8 | 0.93 | 0.92 | 0.76 | 0.74 |
| 8–13 | 0.99 | 0.92 | 0.75 | 0.88 |
| ≥13 | 0.98 | 0.83 | 0.79 | 0.61 |
| All ages combined | 1.0 | 0.92 | 0.87 | 0.73 |

In 2016, we introduced the ADORE system for companion animals that automatically processed and analyzed the data queried from the centralized veterinary database of the participating hospitals (*Kass et al., 2016*). Our overarching goal was to implement the ADORE system in a centralized veterinary medical database for near real-time surveillance for companion animal diseases. Based on our best knowledge, there is currently no surveillance system that can accomplish this for companion animals in the United States. The ADORE system implemented a novel epidemiologic algorithm, the PDOR, and a

**Table 4 Sensitivity analyses of lag time on alert timeline for elective serum creatinine concentration.**

| Lag time | Alert timeline | | |
|---|---|---|---|
| | Start | Peak | End |
| 30-day | Early January 2007 | January 2007 | April 2007 |
| 60-day | Early January 2007 | February 2007 | May 2007 |
| 90-day | Early January 2007 | March 2007 | May 2007 |
| 120-day | Early January 2007 | April 2007 | May 2007 |

scoring matrix based on measures for strength of association and precision (e.g., the PDOR estimate and its corresponding CI) instead of statistical significance. Two simulated outbreaks (i.e., aflatoxicosis and gastrointestinal illness outbreaks) were used to evaluate the performance of the ADORE system and the PDOR procedure during the proof-of-concept phase of the project. The 2007 melamine-related nephrotoxicosis foodborne outbreak in dogs and cats opens up a unique opportunity for us to further evaluate this novel surveillance system. With the documented outbreak record, we were able to specifically test the hypotheses concerning time to detection and outbreak duration. The presumptions of using this historical outbreak were: (1) the causal effect of the chemical adulterants was strong; (2) the prevalence of chemical adulterants in pet food was high; (3) the syndromic components pertaining to nephrotoxicosis were available in the database; and (4) the owners of affected animals sought care at participating veterinary hospitals. Our evaluation using real outbreak data contributes valuable new information by translating theory into a real-life application.

We showed that the performance of the PDOR procedure for detecting this outbreak varied across the selected syndromic components. The alerts generated from the analyses of elevated serum creatinine concentration not only accurately detected the outbreak, but also would have led to detection as early as January 2007, which was 2.5 months earlier than the announcement of the initial pet food recall on March 16, 2007. Furthermore, the patterns of the alerts began with a gradually increasing phase between January and February 2007, followed by a peak from late February to early April, and then a declining phase until early May. These patterns suggest a common source outbreak with a continuous exposure, which accurately aligned with the reported timeline for the outbreak (*U.S. Food & Drug Administration, 2007*). The alerts of elevated creatinine were consistent across species and age groups, except for dogs 13 years old and older, in which the consistent alerts were not observed until late February 2007. These findings comport with our hypotheses that the sensitivity of the PDOR procedure to detect aberrant events could be enhanced by performing analyses stratified by age. Because hypercreatininemia is not commonly reported in healthy young animals (based on baseline windows), this low incidence might have made the PDOR procedure more sensitive to detect an increase in syndromic reporting. In addition, we noted that because the diagnostic outcome proportion (Table 1) takes into account the total number of the medical records, the increased euthanasia of older patients presenting with high serum creatinine concentration
(thus contributing fewer data points) should not greatly affect the sensitivity of the PDOR procedure. Also, if elevated creatinine prompted an owner to immediately euthanize an animal, it would still contribute a record on that day with the corresponding laboratory creatinine value.

Compared to elevated creatinine, the alerts generated from the other syndromic components of clinical presentations (vomiting, anorexia, and lethargy) did not consistently and sensitively detect the outbreak. The majority of the 7-day moving average of the alerts peaked on and after March 21, 2007, which was later than the date of the initial nationwide pet food recall. These discrepancies indicate the potential limitations of using non-specific, owner-reported clinical signs in syndromic surveillance. Because many diseases, pathogens, and toxins can cause these non-specific clinical signs, it might explain the low sensitivity of these syndromic components for detecting this nephrotoxic foodborne outbreak. The late detection might be attributable to the awareness of pet owners about the outbreak, which might have prompted concerned owners to bring their dogs and cats to veterinarians. This was further supported by the increase in the daily number of visits in and after March 2007 (Fig. 1).

There were, on average, 2.2 high-tiered (yellow and higher) alerts per day generated outside the defined outbreak period (i.e., November 1, 2006 to December 2, 2006 and May 1, 2007 to May 31, 2007), compared to 12.6 high-tiered alerts per day generated during the outbreak period (Fig. 3). These findings suggest that the PDOR procedure has high specificity. The PPVs further revealed that elevated serum creatinine outperformed other syndromic components. However, we noted that the lengths of non-outbreak period (62 days; 30%) and outbreak period (147 days; 70%) included in the alert period were not equal, which would have affected PPV estimates. Nonetheless, the PPVs derived from elevated serum creatinine in cats ranged from 0.92 to 1.0, indicating a very low proportion of false alerts. Finally, we stress that these individual alerts would not necessarily trigger an immediate investigation, as the decisions to initiate an investigation must be made by considering medical factors beyond quantitative measures. Thus, a proactive response plan is warranted.

During the investigation, we noticed a lower than expected diagnostic outcome proportion in elevated creatinine from late November 2006 to early January 2007. We confirmed that this was not due to computational errors in data recording, storage, or transfer. A possible explanation was that it occurred during the holiday season (i.e., Thanksgiving, Christmas, and New Year) of that time period. Despite the fact that the system's algorithms would exclude baseline windows with all PDORs <0.8, the sensitivity analyses still revealed the impacts of the low diagnostic outcome proportions during this time period on shifting the peaks of alert scores (Table 1). However, the start dates and duration of alerts were not substantially affected, which indicated that the PDOR procedure was robust for detecting an outbreak with different magnitudes and durations. The use of different lag times did not influence the alert generations from the analyses of vomiting, anorexia, and lethargy. Due to limited time frame of data available for this investigation, we were unable to evaluate the influence of a lag time beyond 120 days and investigate potential seasonality or other long-term periodical variation in alert scores.

However, Figs. 4–7 suggest weekly variation in the studied syndromic components. This supports our use of a 7-day baseline window and presentation of 7-day moving average. Based on the 7-day moving average, no periodic variation in elevated serum creatinine was observed during the study period. In contrast, some periodic variations were observed in other syndromic components. These observations fit with the clinical presentation of the studied syndromic components in the study areas. We do not expect to see a seasonal variation in elevated serum creatinine, but cannot rule this out for other syndromic components. Nevertheless, a longer study period is required to investigate long-term periodical patterns in the alert score of the studied syndromic components.

Although in this post hoc retrospective investigation we selected only a subset of the syndromic components pertaining to the 2007 outbreak of nephrotoxicosis from the original pool, the ultimate objective was to assess the performance of the PDOR algorithm and the ADORE system on prospectively detecting an aberrant event beyond this particular outbreak. The original pool contained 10 syndromic components, including anorexia, elevated alanine aminotransferase, elevated serum calcium, elevated creatinine, diarrhea, lethargy, a Salmonella-positive fecal sample, seizures, urolithiasis, and vomiting. These syndromic components were identified from the database of the participating networked primary care veterinary hospitals for development of a foodborne disease surveillance in companion animals. Thus, it is advised to include all syndromic components for future near-real time foodborne disease surveillance. The findings from this investigation of the 2007 nephrotoxicosis outbreak combined with the evaluations using the simulated aflatoxicosis and gastrointestinal illness outbreaks demonstrate that the PDOR procedure can be applied in syndromic surveillance to effectively and accurately detect various types of foodborne illness outbreaks in companion animals. Additionally, the observed variations in the performance of the PDOR procedure across the ages of animals and syndromic components indicate the importance of integrating veterinary medical knowledge in the utilization and calibration of a surveillance program, such as ADORE, that provides its end users with wide-ranging flexibilities for defining input parameters. Currently, the ADORE system is programmed to integrate the centralized medical database of the participating networked primary care veterinary hospitals in the United States. Nonetheless, the PDOR algorithm can be easily adapted and implemented in other syndromic surveillance programs that utilize clinical diagnostic test results for companion animals.

## CONCLUSIONS

In this study, we evaluated the performance of a novel companion animal syndromic surveillance system for the detection of the 2007 melamine-related nephrotoxicosis foodborne outbreak. With the documented outbreak record, we specifically tested the hypotheses concerning time to detection and outbreak duration. The results showed that the alerts generated from the analysis of elevated serum creatinine concentration could have led to an early detection of the 2007 foodborne outbreak and were well matched to the reported timeline of the outbreak. We also observed variations in the performance of the surveillance system across age of animals and syndromic components, which agreed

with our hypothesis that the sensitivity of detection would be lower in older animals and with non-specific syndromic components due to an expected higher baseline diagnostic outcome proportion. These findings further suggest the importance of integrating veterinary medical knowledge in the utilization and calibration of a surveillance program. Combined with the findings from previous evaluations using simulated outbreak scenarios, this study provided additional evidence that this novel syndromic surveillance system, if implemented in a centralized veterinary medical database, can effectively and accurately detect various types of foodborne illness outbreaks in companion animals.

## ACKNOWLEDGEMENTS

The authors would like to thank Mars Incorporated for providing us veterinary medical data from a centralized hospital database.

### Funding
This project was supported by Mars Petcare, Mars Incorporated, Award # SLO001/UCD Project #200911155. There was no additional external funding received for this study. The funders had no role in study design, data collection and analysis, decision to publish, or preparation of the manuscript.

### Grant Disclosures
The following grant information was disclosed by the authors:
Mars Petcare, Mars Incorporated: Award # SLO001/UCD and Project #200911155.

### Competing Interests
Philip H. Kass is an Academic Editor for PeerJ.

### Author Contributions
- Hsin-Yi Weng conceived and designed the experiments, performed the experiments, analyzed the data, prepared figures and/or tables, authored or reviewed drafts of the paper, and approved the final draft.
- Mark A.L. Gaona performed the experiments, analyzed the data, authored or reviewed drafts of the paper, and approved the final draft.
- Philip H. Kass conceived and designed the experiments, performed the experiments, analyzed the data, authored or reviewed drafts of the paper, and approved the final draft.

### Data Availability
The raw data is available as a Supplemental File.

### Supplemental Information
Supplemental information for this article can be found online at http://dx.doi.org/10.7717/peerj.9093#supplemental-information.

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
