# Peer review of "Evaluation of a novel syndromic surveillance system for the detection of the 2007 melamine-related nephrotoxicosis foodborne outbreak in dogs and cats in the United States"

_PeerJ, doi:10.7717/peerj.9093_

## Round 0.1 · original submission · Minor Revisions

Thank you for your submission to PeerJ. The manuscript has now been reviewed by two experts, and overall the reviewers have commented that the study is interesting, is beneficial for veterinary health informatics research, and should be published.

The reviewers suggest that minor edits will improve the manuscript, including some minor grammatical edits to clarify the language. For example, the use of the acronym PPV should be explained. In the abstract the line reads “during the reported outbreak period (PPV)”, but that reported outbreak period is not the definition of PPV. Please make sure to clarify all acronyms prior to their use. Currently, the manuscript uses quite a few acronyms and comes across as somewhat jargon filled. It might be helpful to reduce if possible.

The authors should make it clear whether or not location was used in the analysis, and if not, why this decision was made. I would imagine that location of sick animals would be relevant in determining outbreaks, and the risk, but location is not a column in the raw data, and it is not explicitly mentioned in the methods section as a variable in the analysis.

Reviewer 1 ·

Basic reporting

I think this is a very valuable paper. It is really well written article and is mostly very clear. There were a few typographic errors and omitted words, I think, possibly cut-and-paste errors. The article was well structured, though the sentence on lines 126-128 is about the benefits of your system rather than the method itself and could be moved elsewhere. You only need to define acronyms once in the abstract and once in the text, rather than in the intro and again in the discussion.

My major source of confusion was the acronym PPV, I presume, is positive predictive value, but PPV is never defined as an acronym. It was sort of defined as an equation, but this was confusing to me: "the time to alert and proportion of alerts generated during the reported outbreak period."

Experimental design

This was a data mining exercise using software to analyze a very large database to determine if specific clinical criteria recorded in medical records could have been used to identify a historic outbreak based on baseline data prior to the outbreak. Though my epidemiology and statistics background is somewhat lacking, I found the study design to be innovative and I could understand the general concept.

Validity of the findings

My statistical background is somewhat lacking, but the tables and charts were easy to follow and very convincing. The results made sense in the context of the historical outbreak in question. Conclusions were well explained and solidly based on the results.

Additional comments

This paper will make a good contribution to the literature, and I am excited about the implications for this software to improve our ability to make timely patient care and diagnostic decisions in the short-term and the potential impact for regulatory agencies in the longer term. My major confusion as a reader who is not an epidemiologist or statistician was on the definition/use of the PPV, which I don't think was well explained or described.

Here are a few specific comments:

Abstract:
L28: What does PPV stand for? Is this the positive predictive value? This was a source of confusion for me throughout the text. Is the PPV the proportion of alerts during the outbreak over all of the alerts? This is unclear to me.

Introduction:
L49: Fraudulently added and adulterated seems redundant in this sentence.
L50: Worth mentioning that the reason a high nitrogen compound was used was to fool the methods used to estimate protein content.

M&M:
126-128: This paragraph describes the benefits of the system, but probably doesn’t belong in the methods section.
L165: A word is missing in this sentence “Because the 2007 outbreak affected across all the US States,” Affected what?

Discussion:
L236 & 290: PDOR and DOP: You don’t have to define acronyms again if you already did so in previous sections.
L278: Garneted isn’t a word. Typo for garnered?

Reviewer 2 ·

Basic reporting

Dates on Figures 4-7 are hard to read, and more-readable ticks could be generated twice a month while still allowing insights related to the key dates of the food intoxication

Line 278, should the word ‘garneted’ be replaced with ‘garnered’

Experimental design

This study elegantly expands on a surveillance system already tested on a simulated disease outbreak. The methodology is applied to a dataset containing an authentic outbreak which is an important step in validating the system.

In terms of replicating data, the author could clarify how records are used. As a reader, I assumed that records were processed on a per record basis not a per-animal basis. As a consequence, one animal might contribute multiple entries to the PDOR calculation. So, in Figure 1, Is one record a summary of all data collected about one animal on one day ?

Validity of the findings

The data suggests that the system could be used to detect food-borne disease outbreaks – as long as it is running in real time – is the system being used as part of an on-going surveillance system ?

Given that abnormal creatinine levels were the key driver in identifying the outbreak, can the authors comment whether a simpler monitoring of proportion of cases identified with high creatinine set against a simple threshold would have performed worse

The authors discuss the differing findings associated with age when assessing the creatinine-derived signal. The conclusion that this highlights the value of stratifying data by age to improve sensitivity. The data might also reflect the increased euthanasia of older patients presenting with high creatinine which would also decrease the signal if records are handled on a per-record basis (euthanased animals contributing fewer data points and therefore a shorter signal)

The authors do not comment on the potential biphasic nature of the data initial rise early January, second rise mid February – potentially highlighted by the older dog data – was there any information about batch-releases of contaminated foods that might corroborate this ?

Additional comments

This is really interesting work and helps to build the toolkit for veterinary health informatics research.

---

## Round 0.2 · Minor Revisions

Thank you for your Revision. I am recommending minor revisions again because not all the reviewer comments were addressed directly in the manuscript. The manuscript is very well written, clear and easy to understand. It is clear that a lot of time and effort has been made in the development of this study and manuscript. However, it is important to respond to reviewer comments in the rebuttal letter, but also make appropriate clarifying changes directly in the manuscript to help improve the readability. For example, Reviewer 2 made a number of relatively simple to address comments that, if addressed, could improve the functionality and broad-ranging usefulness of the ADORE system as presented in the manuscript. However, those implications have not been adequately addressed in the manuscript itself.

Reviewer 2, comment 3: Please define what you mean by records in the methods, and again in the figure legend.

Reviewer 2 comment 4: Please mention in the manuscript itself what you have mentioned in the rebuttal letter. As you say in the 1st paragraph of the discussion, there is a lack of centralized vet medical database, yet you did have access to one for this study, and mention that you introduced the ADORE system – is it not yet widely available? If relevant, within the manuscript you should call for changes that will improve the system. Lines 322-324 begin to make the case for the utilization of a surveillance program but fall short because it is not described how this might work.

Reviewer 2 comment 5: This comment was not addressed in the manuscript. Creatinine level was certainly important for this analysis – and claimed in line 188 that it is the best predictor for an outbreak, as well as the claims made in as abstract lines 32-34. The reviewer comment has merit: would it be adequate to monitor more simply for elevated creatinine levels? The authors move from talking directly about the specific outbreak, to a quick switch to 10 syndromic compounds for detection in paragraph starting on line 311. It might benefit from a paragraph reordering, with edited lines 316-320 coming before line 311.
Please discuss more in more depth why a shorter list of syndromic components would not be sufficient. Further, the rebuttal mentions clear edits of line 316-324, but according to the tracked changes document, no clear changes have been made to address this reviewer comment.

Please address Reviewer 2 comment 6 directly in the manuscript.

---

## Round 0.3 · accepted · Accept

Thank you for submitting your revised manuscript. These additional edits have improved the clarify and highlighted the importance of the study. Congratulations on the paper's acceptance! I'm excited to see it published.